# Dolphin Watching and Compliance to Guidelines Affect Spinner Dolphins’ (*Stenella longirostris*) Behaviour in Reunion Island

**DOI:** 10.3390/ani11092674

**Published:** 2021-09-12

**Authors:** Belén Quintana Martín-Montalvo, Ludovic Hoarau, Ophélie Deffes, Sylvain Delaspre, Fabienne Delfour, Anne-Emmanuelle Landes

**Affiliations:** 1Centre d’Etude et Découverte des Tortues Marines (CEDTM), 97424 Piton Saint Leu, France; ophelie.dfs@hotmail.fr (O.D.); sylvaindelaspre@cedtm-asso.org (S.D.); aelandes@cedtm-asso.org (A.-E.L.); 2UMR ENTROPIE, Université de La Réunion, IRD, CNRS, IFREMER, UNC, CEDEX 9, 97744 Saint Denis, France; ludovic.remy.hoarau@gmail.com; 3Faculté des Sciences et Techniques, Université Jean Monnet, 42100 Saint-Etienne, France; 4Ecole Pratique des Hautes Etudes, CRIOBE, 75014 Paris, France; fabienne_delfour@yahoo.com

**Keywords:** behavioural responses, human-wildlife interactions, wild dolphin welfare, conservation, animal-based measures, management, Reunion Island

## Abstract

**Simple Summary:**

Whale and dolphin watching have expanded worldwide, with their impacts on cetaceans over the short and long terms being widely reported. In Reunion Island, the activity has rapidly developed, notably around a resident spinner dolphin population, which can be seen year-round. Focal follows revealed that the dolphins are less likely to begin socialising or resting and more likely to remain travelling or milling in the presence of dolphin-watching vessels. The dolphins’ avoidance reactions increased when tourist vessels were numerous or in non-compliance with the regulations. The cumulative effect of such behavioural responses likely impacts the dolphins at the population level, highlighting the need for enforcing adaptive and efficient dolphin-watching management measures to ensure the welfare and preservation of the resident spinner dolphins of Reunion Island.

**Abstract:**

Marine wildlife tourism has rapidly developed in Reunion Island, due to a large demand for interactions with a resident population of spinner dolphins (*Stenella longirostris*). The presence of dolphin-watching vessels close to dolphin groups can cause short-term impacts on their behaviour; cumulative incidences likely result in deleterious long-term impacts on the population. Using scan sampling, we collected behavioural data on spinner dolphins to evaluate the short-term effects of dolphin watching on their behaviour. The dolphins were less likely to begin socialising or diving while travelling and more likely to stay travelling and milling in the presence of vessels. Additionally, activity budgets for resting and socialising decreased simultaneously with increased travelling and milling. Avoidance responses significantly increased with an increase in the number of vessels and non-compliance with the local dolphin-watching guidelines. These behavioural responses are likely to have energetic costs for the dolphins, which may lead to decreased survival and reproductive success at individual and population levels. More restrictive regulations, increased surveillance and animal-based measures are key tools to adapt the conservation efforts in Reunion Island. Further studies on the island’s resident dolphin populations are necessary to ensure the continuity of dolphin-watching activities in a sustainable manner.

## 1. Introduction

Interest in observing and interacting with cetaceans in their natural environment has become a major recreational activity and industry worldwide [1,2,3]. Whale or dolphin watching is defined as observing cetaceans in their natural environment, from a nautical, land or aerial base, as well as any practice of swimming with cetaceans [4]. Cetacean watching can serve as a useful tool to raise public awareness of the importance of cetaceans and their conservation, whilst supporting sustainable nature-based tourism [5,6]. However, some cetacean-watching practices can impact the welfare of the targeted species, compromising the sustainability of the activity. Therefore, it is necessary to document and increase knowledge of its impacts [3,7,8,9]. Identifying and quantifying the impacts of cetacean watching have become a growing concern for conservation management and are the prerequisite to actioning a reduction in any short- and long-term consequences it may have on cetaceans [10,11,12]. Changes in cetacean behavioural states are easily measured in the field and are frequently used to evaluate short-term disturbances resulting from dolphin-watching vessels on the targeted species [11,12,13,14,15,16]. Dolphins react to the vessels’ presence by showing short-term antipredator behavioural responses, such as forming tighter groups [10,17,18], altering their swimming speed and direction [18] or displaying more erratic surface movements [13,15,19]. The reaction to the vessels’ presence may also be dependent on the groups’ size and cohesion: studies have found that tighter groups show increased vigilance compared to dispersed groups, and smaller groups show more avoidance reactions than larger ones [12,20]. Dolphins often display avoidance reactions before a vessel arrives, as they can detect the underwater noise produced by engines from a distance [18,19,21,22,23]. Avoidance responses can include increasing the depth and duration of dives [18,19,21,24] and reducing socialisation such as resting and feeding activities with conspecifics in favour of travelling [11,14,16,20]. Such alterations in behavioural budgets have great energetic costs for the dolphins, increasing their physical demand [11,13,19,21], which over the long term can contribute to reduced reproductive success and population decline [10,21,25,26,27]. The cumulative short-term negative effects caused by repeated interactions with anthropogenic activities may also lead to a total avoidance of certain areas [10,15,28,29]. During the last decade, cetacean watching has greatly developed in Reunion Island, a French oversea territory in the south-western Indian Ocean [30,31]. Dolphin watching was historically the first cetacean-watching activity regularly practised on the island [30]. The record presence of humpback whales (*Megaptera novaeangliae*) on its coasts during the 2017 and 2018 southern winters for breeding [32] catalysed the tourism offer, and cetacean watching is today a major local activity. To date, both commercial and recreational activities take place in Reunion Island, with a fleet of, respectively, 50 and 30 vessels with maximum passenger capacities ranging from 10 to 80 (personal observation). The decrease in the whales’ occurrence in the 2019 and 2020 breeding seasons, the well-established cetacean-watching commercial offer and the high demand from both foreign visitors and local consumers have meant a redirection of the attraction and commercial offer to coastal dolphin populations [33,34]. A resident population of spinner dolphins (*Stenella longirostris*, Gray 1828) is commonly observed throughout the year on the west coast of the island [32,33,34,35]. Estimated at 175 individuals, this population is potentially genetically distinct from other spinner dolphin populations in the south-western Indian Ocean [36]. Its core habitat lays between 45 and 75 m depth in front of Saint-Gilles Port, where most of the maritime and touristic activities occur [35]. Spinner dolphins typically spend the morning resting and socialising in shallow waters close to the coast before travelling offshore to forage at night [25,37,38]. The particular ecology of spinner dolphins, with predictable daily migrations across spatially limited areas, facilitates dolphin-watching vessels to observe them [25,26,37,39,40,41]. Due to the easiness of encounters, especially in the morning, the population of spinner dolphins is the most consistent main target of the cetacean-watching activities in Reunion Island [31]. The vulnerability of this population [35] calls for a deeper insight into this increasing trend in human–animal interactions. In Reunion Island, a voluntary code of conduct has been implemented since 2009 to ensure respectful practices during cetacean-watching experiences [42]. In 2019, the local government enforced legislation to legally support the code of conduct at a regional level. This legislation was updated in July 2020 with more restrictive guidelines [43]. The recent regulations comprise both hard (prefectural decree) and soft (code of conduct) laws, contributing to a fragmented legal framework at the regional scale and discontinuous protection of cetaceans from one territory to another within the western Indian Ocean [44]. While several studies exist on the ecology and distribution of the dolphin populations of Reunion Island [32,35,36], few have focused on the impacts of cetacean watching, and those few studies have concentrated on impacts on humpback whales [45,46]. This study aims to assess for the first time the effects of dolphin watching on spinner dolphins’ behaviour through a focal follow instantaneous scan sampling approach. This allows an assessment of which factors could affect their welfare and conservation over the long term, and thus help to establish animal-considered measures that ensure the sustainability of the dolphin-watching industry in Reunion Island.

## 2. Materials and Methods

### 2.1. Study Area

Data were collected between February 2018 and June 2020 on the western coast of Reunion Island (55°33′ E, 21°07′ S), up to 6 nautical miles offshore and only with sea conditions equal to or under 3 (Beaufort scale). All surveys were conducted between 7 a.m. and 2 p.m. The western coast in front of Saint-Gilles Port was defined as the study area because it contains overlaps of both the core spinner dolphin habitat [35] and the majority of maritime activities, including cetacean watching (Figure 1).

### 2.2. Data Collection

Surveys were conducted by researchers of the Centre d’Etude et de Découverte des Tortues Marines from a semi-rigid research vessel (5.7 m long, 70 cv horsepower). When a dolphins’ group was sighted, the research vessel was carefully positioned at a 300 m radius from the outermost edge of the group. The vessel was kept parallel to the dolphins’ swimming direction to allow for an overall view of the group. Dolphin-watching vessels are typically positioned at a distance between 50 and 300 m from the animals, in accordance with the local regulations [43]. The vessels at a 300 m radius from the group are deemed to have a lower impact on the dolphins than that of the closer vessels. Minimal changes in speed and gearshifts were ensured to limit disturbance to dolphins. Maintaining such an approach protocol minimised the impact of the research vessel on the dolphins, especially compared to that of dolphin-watching vessels [13,14].

#### 2.2.1. Group Composition and Behavioural State

Instantaneous scan sampling protocols were employed at 3 min intervals to record the predominant group behaviour of the majority (≥50%) of individuals in the focal group and dolphin group location [47,48]. Instantaneous scan sampling was conducted for a minimum of 6 min (2 scans) and for a maximum of 45 min (15 scans) in accordance with the local observation guidelines or as soon as visual contact with the group was lost. Group size (i.e., number of dolphins within the group) and composition (i.e., adults and calves) were estimated by the naked eye, which can lead to an over- or under-estimation of the group size. A calf is defined as an individual distinctly smaller, usually half the size, than the adult next to which it swims [49]. Group cohesion was noted as high (dolphins in the focal group were highly aggregated occupying a small area, usually smaller than 30 m), intermediate (dolphins occupied an area approximately between 50 and 100 m wide) and low (dolphins were scattered through a sometimes large area, more than 100 m wide) [50]. Derived from definitions used in other dolphin impact assessment studies, the behavioural states were categorised as resting, travelling, socialising, diving, milling and feeding [11,49,51,52,53,54]. All behaviours were mutually exclusive. Prior to data collection, two researchers were trained in the field in 2017 and in early 2018 to ensure reliability of data collected between the two observers and to identify the dolphins’ behaviour as defined in the ethogram. The same observers collected the data throughout the study. No feeding behaviours were observed during the study (Table 1).

#### 2.2.2. Dolphin Behavioural Response to Vessels

The total number of vessels present within a 300 m radius from the group and the predominant group behaviour displayed in response were recorded. Responses were defined as (i) avoidance, when the group changed its course or direction, disappeared from the observation area or abruptly changed its behaviour; (ii) neutral, when no particular reaction was displayed or no brutal changes in behaviour or course were adopted; and (iii) attraction, when the group explicitly approached or interacted with the vessels, also called “bow riding”.

#### 2.2.3. Vessels’ Compliance with the Regulations

A dolphin-watching vessel was noted as non-compliant when at least one recommendation of the regulation was not respected. Non-compliance with the guidelines includes, amongst others, exceeding the observation time (45 min when there are less than five vessels, 15 min when more than five), exceeding the number of vessels within the 50 to 300 m radius from the group (five vessels), approaching the dolphins at a closer distance than 50 m and crossing the dolphins’ path or encircling them. All parameters were estimated by the naked eye by the two researchers previously trained in the field from 2017 to early 2018 (i.e., visualization of known distances between two points in the sea). However, this can lead to miscalculations of time, distance and trajectories, whereas the number of vessels in the observation area is more accurately reported.

### 2.3. Data Analysis

#### 2.3.1. Markov Chain

First-order time-discrete Markov chains were used to study the probabilities of passing from one behavioural state to the 3 min succeeding state under control and impact conditions [11,13,55,56]. The control conditions were met when no dolphin-watching vessels were present in the observation area. The impact conditions were met when at least one dolphin-watching vessel was present between two behavioural states. The research vessel was assumed to have a negligible impact on the dolphins’ behaviour and therefore not included in the analyses. When a single impact condition was followed by a control condition or vice versa, it was noted as a “transition”. Transition conditions are not considered as either impact nor control, and they were thus excluded from the analyses [13]. The impact and control contingency tables were compared using chi-square tests. All analyses were performed in R Studio (version 3.6) using the markovchain and poLCA packages [57,58].

##### Behavioural Transitions

Transition probabilities between preceding and succeeding behavioural states were obtained from the contingency tables for the control and impact chains separately [13]:$$p_{ij}=\frac{a_{ij}}{\sum_{i=1}^{n}a_{ij}}\;,\;\sum_{i=1}^{n}p_{ij}=1$$
where *i* is the preceding behavioural state, *j* is the succeeding behavioural state, *a**_ij_* is the number of transitions observed from behavioural states *i* to *j*, *p**_ij_* is the transition probability from *i* to *j* in the Markov chain and *n* is the total number of behavioural states (i.e., five states).

##### Behavioural Budget

The behavioural budget, defined as the difference in the proportion of time spent in each behavioural state between control and impact conditions, was tested using the chi-square test. Two-sample tests for equality of proportions were used to compare each behavioural budget between control and impact conditions.

##### Bout Length

The mean duration of the time spent by the dolphins in the same behavioural state *t_ii_* was calculated:$$t_{ii}=\frac{1}{1-p_{ii}}$$
with a standard error of $SE=\sqrt{\frac{p_{ii}\times \left(1-p_{ii}\right)}{n_{i}}}$, where *n**_i_* is the number of samples with *i* as preceding behaviour. Student’s *t*-tests were used to compare bout lengths between control and impact conditions [13].

##### Recovery Time

The recovery time, or the number of transition units it took the spinner dolphins to return to each behavioural state, was estimated for control and impact conditions [52]:$$E\left(T_{j}\right)=\frac{1}{\pi_{j}}$$
where (*T**_j_*) is the time (i.e., number of transitions multiplied by the length of each transition unit, that is, 3 min) it takes the spinner dolphins to return to state *j* (behavioural state they currently were in) and *π* is the steady-state probability of each behaviour in the chain. The recovery times in control and impact conditions were then compared; high values indicated longer recovery times for dolphins to return to each initial state, and low values indicated short recovery times.

#### 2.3.2. Mixed-Model Analyses

Generalised linear mixed-effects models (GLMMs) were performed to evaluate the dolphin groups’ behavioural response to vessel presence and compliance using the R package lme4 [59,60]. Dolphins’ responses were treated as binomial, i.e., non-avoidance (approach or no reaction of the group) vs. avoidance. The group response observations (n = 320) were not independent, because some groups had been resampled, and therefore focal group was included as a random effect. Candidate models included combinations of five variables that potentially affected the dolphin group response: group behaviour (see Table 1), vessels (number of vessels present around the dolphin group), non-compliance/compliance (whether vessels were compliant with the regulations; see Section 2.2.3), cohesion (i.e., low; intermediate; high) and group size (corresponding to a visual estimate of the number of individuals in the focal group). Interactions between explanatory variables were not considered. Models were compared using the Akaike information criterion corrected for small sample sizes (AIC_c_). The model with the lowest AIC_c_ was the most parsimonious one from the model set [61]. To evaluate the model fits, the diagnostic tools provided by the DHARMa package in R [62] were used. Model diagnostics were performed by creating scaled residual simulations from the fitted model with the function simulateResiduals (number of simulations: 1000). The residuals were plotted against the predicted response from the model by using the function plotSimulatedResiduals.

## 3. Results

### 3.1. Effects of Dolphin-Watching Vessels on Spinner Dolphins’ Behaviour

Between February 2018 and June 2020, we encountered spinner dolphin groups on 57 occasions on the west coast of Reunion Island. The dolphins were accompanied by dolphin-watching vessels at the time of the first sighting in 95% of the cases (n = 54 groups). Behavioural data were collected over 24 focal group follows (Table 2), corresponding to a total of 15.75 h of behavioural data. During this time period, 342 behavioural transitions were recorded, of which 70 (20.5%) were classified as control and 272 (79.5%) as i. Control sequences lasted an average of 25 min (±SD = 20 min), and impact sequences averaged 35 min (±SD = 21 min). Only one group was followed per day (mean group size = 50.87 ± 14.63 dolphins). Behavioural data were mainly collected for adult individuals, as adults represented the majority (≥50%) of individuals in every focal group. No group fusion or fission situations were recorded.

#### 3.1.1. Behavioural Transitions

The results of Markov chain analyses showed a significant effect of dolphin-watching vessels on the transition probability between behavioural states (goodness-of-fit test: χ^2^ = 77.713, df = 16, *p* < 0.001; Figure 2). In the presence of dolphin-watching vessels, the probability of passing from travelling to socialising and to diving decreased, respectively, from 0.50 to 0.07 (χ^2^ = 3.7016, df = 1, *p* = 0.05) and from 0.50 to 0.04 (χ^2^ = 6.6762, df = 1, *p* < 0.01). Moreover, the transition probability of remaining travelling and milling increased, respectively, from 0.04 to 0.50 (χ^2^ = 10.784, df = 1, *p* < 0.01) and 0.58 to 0.95 (χ^2^ = 7.7381, df = 1, *p* < 0.01), while the probability of remaining diving increased, although not significantly.

#### 3.1.2. Behavioural Budget

The spinner dolphins’ behavioural budget was significantly affected by the presence of vessels (χ^2^ = 26.321, df = 4, *p* < 0.001; Figure 3). The groups spent significantly less time socialising in impact (10%) than in control conditions (33%; χ^2^ = 21.666, df = 1, *p* < 0.001). A similar trend was described for resting (respectively, 27% and 42%; χ^2^ = 4.9902, df = 1, *p* < 0.05). Inversely, dolphin groups spent significantly more time travelling (23% to 8%, χ^2^ = 6.7106, df = 1, *p* < 0.05) and milling (32% to 7%, χ^2^ = 15.726, df = 1, *p* < 0.001) when vessels were present.

#### 3.1.3. Bout Length

The average bout length of diving increased from 3 min to 5 min when dolphin-watching vessels were present (Student’s *t*-test, *t* = −2.907, df = 23, *p* < 0.05). In impact conditions, there was an increase in the average bout lengths of milling (from 7 min to 60 min, Student’s *t*-test, *t* = −175.695, df = 50, *p* < 0.001) and travelling (from 3 min to 16 min, Student’s t-test, *t* = −175.695, df = 50, *p* < 0.001) in comparison with control conditions. Conversely, the duration of socialising bouts was greater in control (13 min) than in impact (10 min) conditions (Student’s *t*-test, *t* = 6.084, df = 67, *p* < 0.001). The resting bout length did not significantly vary in the presence or absence of vessels (17 min, Student’s *t*-test, *t* = 0.412, df = 91, *p* = 0.68; Figure 4).

#### 3.1.4. Recovery Time

Recovery times were also altered in the presence of dolphin-watching vessels. The spinner dolphins, in control conditions compared to impact conditions, took longer to return to socialising (9 min in control against 29 min in impact conditions), resting (7 min in control against 10 min in impact conditions) and diving (38 min in control against 50 min in impact conditions). In contrast, the recovery times for travelling (34 min in control to 12 min in impact conditions) and milling (39 min in control to 9 min in impact conditions) decreased in the presence of dolphin-watching vessels.

### 3.2. Compliance of Dolphin-Watching Vessels and Dolphins’ Response

Dolphin-watching vessels were compliant with the recommendations in force in 86.6% of the scans (n = 220) and non-compliant in 13.4% of the scans (n = 34). The number of dolphin-watching vessels interacting with a dolphin group ranged between 0 and 12 per sighting, with an average of two vessels (2.4 ± 1.8, range: 0–12, n = 212 scans). The best-fit model, carrying 76% of the cumulative model weight, included every parameter except group behaviour. The GLMM with the most support for avoidance responses of the dolphin group included the variables vessels, compliance, group size and cohesion (Table 3). Higher spinner dolphin avoidance probabilities were significantly and positively associated with the number of dolphin-watching vessels. Likewise, higher avoidance responses were displayed when vessels were non-compliant with the guidelines (Figure 5). Avoidance probability also increased with group size and was the highest for intermediate group cohesion.

## 4. Discussion

This study showed that dolphin-watching vessels alter the behaviour and activity budgets of spinner dolphins. The dolphins appeared to spend less time resting and socialising, while they invested more time in travelling and milling. In addition, the avoidance responses of the dolphins were significantly affected by the number of vessels and by non-compliance with the regulations of Reunion Island. These findings have important implications for the welfare and conservation of the investigated population of spinner dolphins.

Results revealed that the behavioural budgets for resting and socialising are greater in the absence of vessels, suggesting that when cetacean-watching encounters begin, the dolphins are presumably in those behavioural states. Spinner dolphins’ daily behaviours are well documented [22,63], although survey efforts may not have encountered periods where other behaviours could have been displayed in the absence of vessels, such as feeding behaviours, which were absent in this study. However, the decreased probability of remaining resting or socialising in the presence of vessels demonstrates the short-term impacts of vessels on the observed spinner dolphins’ behaviour. The differences in the recovery times for these behaviours suggest an interruption provoked by the dolphin-watching vessels. In a similar manner, in the absence of vessels, socialising bout lengths were greater than when vessels were present, as already reported in previous studies [13,16]. Socialising and resting are fundamental activities for dolphins [38], and their disruption by dolphin-watching vessels may induce physiological stress and, in the long term, negatively affect the survival and fitness of a population [11,15,21,64]. Indeed, the repeated disturbance by vessels can decrease the rate of successful mating attempts by interrupting the socialisation states [19,65] or harming the cooperation and social cohesion of groups [15,19,66].

In Reunion Island, as a first step to minimise disturbance during critical periods for the species, the decree amended in July 2020 established a quiet period from 6 p.m. to 9 a.m., during which no cetacean species can be approached closer than 300 m [43]. For other dolphin species, spatio-temporal mitigation measures to reduce the pressure during vital periods have been widely suggested as an effective management tool [9,14,18,21,37,67]. This study was, however, carried out before the implementation of this period of quietness. Measuring animal welfare is a common phenomenon, especially in captivity, that is assessed using multidimensional resources and animal-based measures [68]. Consequently, further research on the daily movements, reproductive seasonality and habitat use of the island’s spinner dolphin population [3,15,25,63,69], as well as individual physiological parameters (e.g., faecal glucocorticoid metabolites [70]), is necessary to inform the regulation amendments and effectively manage the welfare of Reunion Island’s spinner dolphin population.

In the presence of dolphin-watching vessels, spinner dolphins were significantly more inclined to begin travelling and milling than socialising or resting. These short-term changes in behaviour have commonly been reported in previous studies as a direct avoidance strategy towards approaching vessels [11,13,15,16,71,72], which culminated in higher behavioural budget allocation for travelling and milling when vessels were present. Similar trends were reported in spinner dolphins in Hawaii [12,22,63] and Egypt [37] and in common bottlenose dolphins in Panama [16] and New Zealand [14]. In the presence of vessels, the recovery times for these behaviours are smaller, suggesting that when disturbed, dolphins tend to favour these activities. Travelling behaviour is an immediate response but costly in energy [13,14,16], whilst milling is thought to be used by the dolphin groups for decision making, to communicate and determine whether to avoid a disturbance and reduce the energetic cost [73,74,75]. The trade-off between travelling and socialising behaviours may also depend on the absence of other suitable habitats [76,77] and on the energetic capability of individuals to move [76].

Spinner dolphins were more likely to avoid vessels when they were not compliant with the enforce guidelines, particularly when the number of vessels was high, in accordance with previous studies on cetaceans [3,19,20,22,37,72]. Although the Reunionese regulation authorises a maximum of 5 vessels in a cetacean observation zone, up to 12 vessels were reported during this study. Interestingly, based on the best-fit model, there is a 50% chance for an avoidance response of the dolphin group when four vessels are present in the observation area, suggesting the current regulations would need to be adapted to minimise the impact on spinner dolphins. Limiting the number of vessels per encounter or per tour operator is considered an effective strategy to reduce disturbance to cetaceans [9,16,55,78,79]. Interactions remain, however, stressful for the dolphins, even if they do not directly avoid the vessels [17]. Increasing surveillance in the dolphin observation areas or establishing sanctioning systems could be a strategy to ensure best practices [3,80]. As seen in this study, when vessels comply with the regulations, chances of avoidance reactions from the dolphins substantially decrease. Improved coordination and cooperation between tour operators during sightings could facilitate respect for the guidelines [9,80], such as, in the Reunionese context, the approach, the disposition and the intention of manoeuvre of the vessels in observation.

Although results indicate that the spinner dolphins’ behaviour was directly impacted by dolphin-watching vessels, a possible misinterpretation of such responses by the researchers cannot be neglected [27,74]. An indirect impact on dolphins’ behaviour, not considered in this study, could be the engine noise produced by the vessels [22,27]. Dolphins can change their behaviour before the arrival of vessels as soon as they detect the noise [19], and different noise levels can make dolphins react in different manners, including the one produced by the research vessel. Besides, dolphin behaviours were interpreted without considering potential external factors that may have influenced them, due to the lack of such data in the area. Field difficulties were encountered, such as the low number of focal follows in control conditions and visual impediments for data collection due to the presence of numerous vessels moving around the dolphins.

Finally, the particularity of the whale- and dolphin-watching activity in Reunion Island is that approximately half of the vessels are for recreational uses (e.g., rented [31,46]). Encouraging education and awareness-raising campaigns is essential to inform the local communities and tour operators about the impacts of the activity, the best practices and the changes in the regulations [9,21,46,56,81]. Nonetheless, in Reunion Island, underwater interactions also take place [34], whose short-term effects have been widely reported worldwide [20,22,51,82]. Although swim-with-dolphin activities were not assessed in the scope of this study, it would be interesting to conduct further research on this practice and its impact on the spinner dolphins’ behaviour. The outcomes of such studies must assist decision makers in adapting conservation regulations to support the welfare of the wild populations targeted by the cetacean-watching industry in Reunion Island [15,25,69].

## 5. Conclusions

In the recent years, the rapid increase of dolphin watching at a global scale has raised concerns about its sustainability and research to evaluate its impacts on cetaceans is expanding. In Reunion Island, this activity has rapidly developed in the last decade and includes commercial and recreational users. This study is the first to assess the behavioural responses of spinner dolphins to dolphin-watching vessels in Reunion Island. The resident spinner dolphin population showed short-term antipredator behavioural responses to the vessels’ presence, as seen in other studies around the world. They spent less time resting and socialising and more time travelling and milling, which can have, in the long run, dramatic consequences for the welfare and conservation of the species. When dolphin-watching vessels were too numerous and if vessels did not comply with the regulations in force, dolphins responded with higher avoidance reactions. The growth of dolphin watching in Reunion Island and its impact on the behaviour of spinner dolphins demonstrate a need to adapt the current management measures to ensure dolphin welfare and the development of sustainable marine wildlife tourism.

## Figures and Tables

**Figure 1 animals-11-02674-f001:**
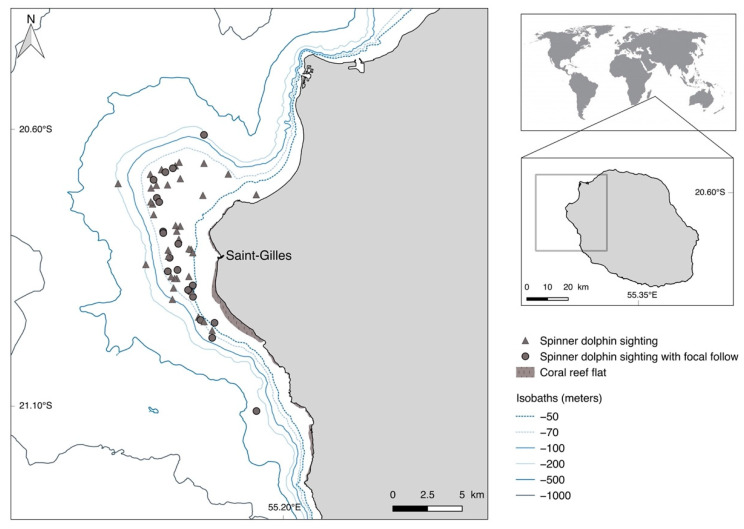
Study area and spinner dolphin sightings between February 2018 and June 2020 along the western coast of Reunion Island in the south-western Indian Ocean.

**Figure 2 animals-11-02674-f002:**
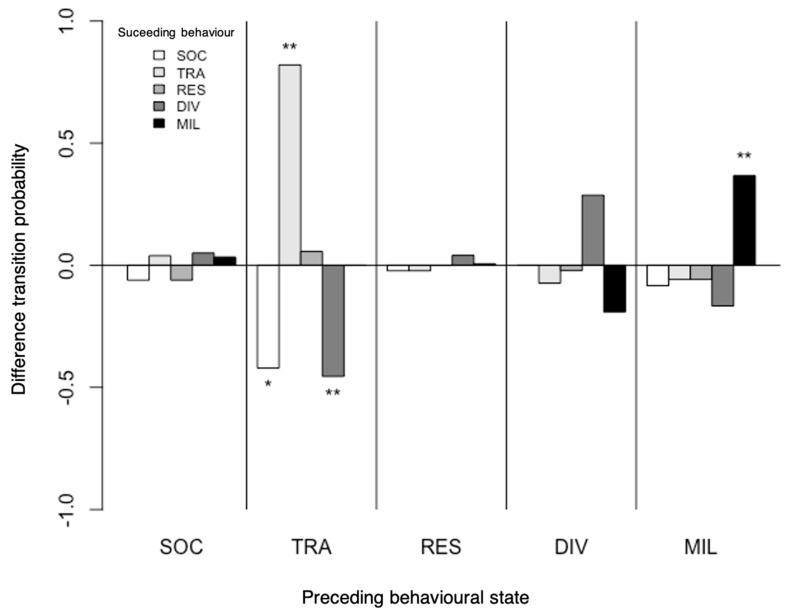
Difference between the transition probabilities in the behavioural states of spinner dolphins in the presence (impact) and absence (control) of vessels. A higher transition probability in impact conditions compared to control ones translates into positive values in the *y*-axis. Significant differences are indicated with an asterisk (* *p* < 0.05, ** *p* < 0.01). SOC: socialising; TRA: travelling; RES: resting; DIV: diving; MIL: milling.

**Figure 3 animals-11-02674-f003:**
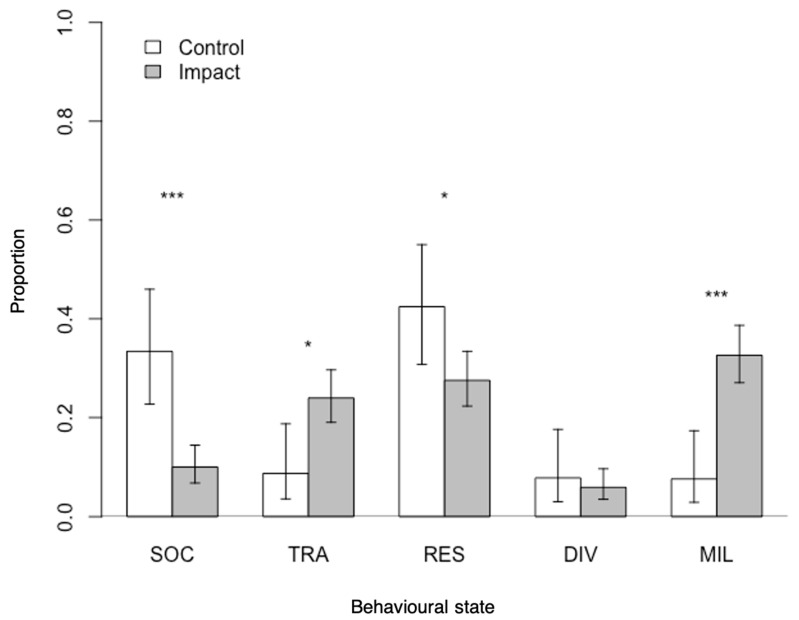
Behavioural budget proportions of spinner dolphins in the presence (impact) and absence (control) of dolphin-watching vessels. Error bars represent 95% confidence intervals. Significant differences are indicated with an asterisk (* *p* < 0.05, *** *p* < 0.001). SOC: socialising; TRA: travelling; RES: resting; DIV: diving; MIL: milling.

**Figure 4 animals-11-02674-f004:**
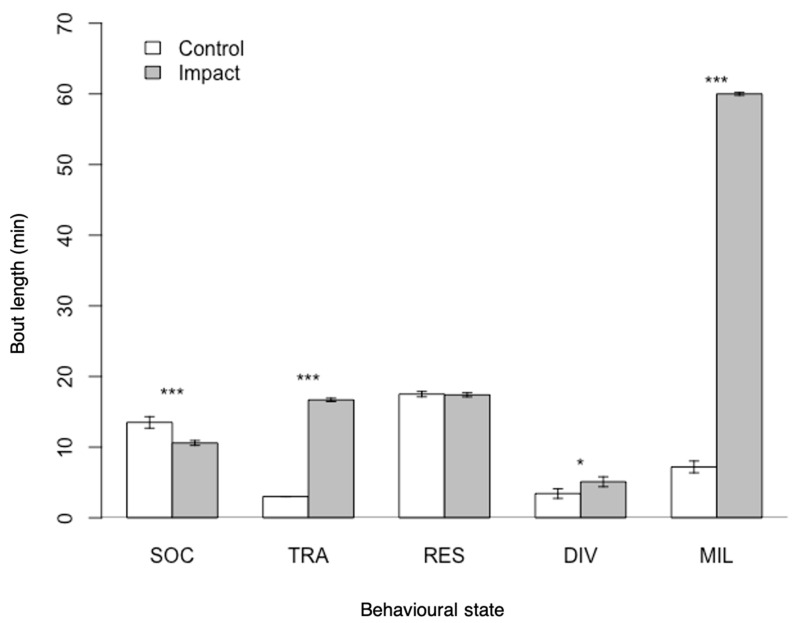
Average bout lengths of the behavioural state of spinner dolphins in the presence (impact) and absence (control) of dolphin-watching vessels. Error bars represent 95% confidence intervals. Significant differences are indicated with an asterisk (* *p* < 0.05, *** *p* < 0.001). SOC: socialising; TRA: travelling; RES: resting; DIV: diving; MIL: milling.

**Figure 5 animals-11-02674-f005:**
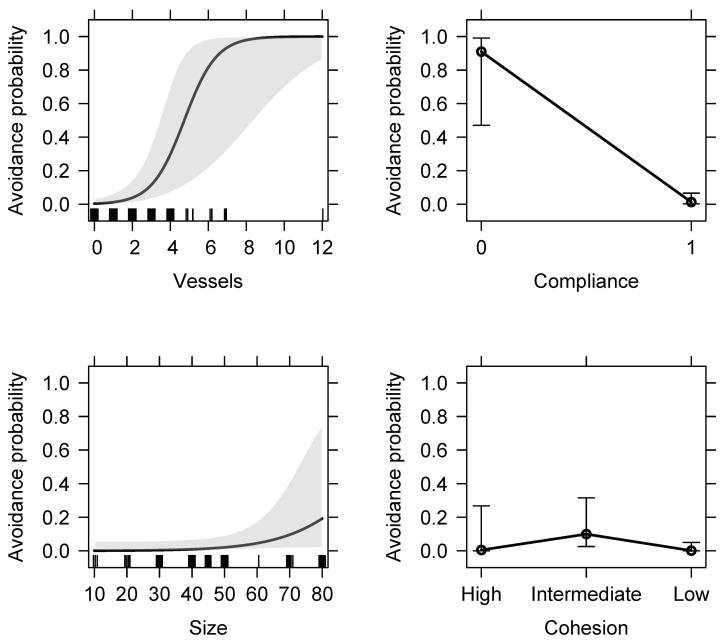
GLMM-predicted probabilities of avoidance of spinner dolphin groups in Reunion Island during vessel approaches, as predicted by the best generalised linear mixed-effects model, which included the number of vessels around the group (top left), non-compliance (=0) and compliance (=1) of vessels with the regulations (top right) and dolphin group size (bottom left) and cohesion (high, intermediate, low) (bottom right). The grey areas and error bars represent 95% confidence intervals.

**Table 1 animals-11-02674-t001:** Ethogram of the spinner dolphins’ behavioural states recorded during the focal group follows. Adapted from Shane [49], Stensland and Berggren [51], Stockin et al. [52], Norris and Dohl [53], Norris et al. [54] and Christiansen et al. [11].

Behavioural State	Definition	Spinner Dolphin Groups during the Study (Credits: CEDTM, A. Nguyen)
Resting	The dolphins stayed close to the surface and close to each other. They surfaced at regular intervals in a coordinated way, either not propelling themselves at all or moving slowly.	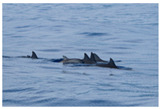
Travelling	The dolphins were propelling themselves at a sustained speed (>3 knots), all heading in the same direction and advancing perceptibly along a certain course.	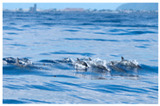
Socialising	This category covers all physical interactions that take place between members of a group, including chasing, body contact and copulation. Socialisation is often accompanied by aerial behaviour.	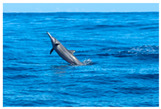
Diving	The group of dolphins was seen diving synchronously for several minutes and disappearing from the view of the observers at the surface of the water. This behaviour is possibly related to feeding or avoidance behaviour.	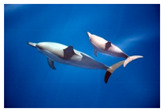
Milling	The dolphins were swimming, but frequent changes in direction prevented them from making any discernible progress in any direction, and so they remained in the same general area. Often, different individuals in the group would swim in different directions at any given time, but their frequent changes in direction kept them together.	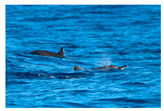

**Table 2 animals-11-02674-t002:** Details on the focal follows carried out between February 2018 and June 2020 on spinner dolphin groups.

Year	Measured Indicator of the Focal Follow	2018	2019	2020	Total
Number of sightings		23	27	7	57
Number of focal follows		3	13	8	24
Impact conditions	Number of behavioural transitions	39	156	77	272
	Total duration of group follows (HH:MM:SS)	01:52:00	06:38:31	03:46:01	12:49:00
	Mean duration of group follows (±SD) (HH:MM:SS)	00:37:20 (±00:15:22)	00:33:13 (±00:26:42)	00:37:40 (±00:10:24)	00:34:59 (±00:21:27)
Control conditions	Number of behavioural transitions	0	11	59	70
	Total duration of group follows (HH:MM:SS)	0	00:18:53	2:38:05	02:56:00
	Mean duration of group follows (±SD) (HH:MM:SS)	0	00:06:18 (±00:06:25)	00:39:31 (±00:12:20)	00:25:17 (±00:20:08)

**Table 3 animals-11-02674-t003:** Model selection for the generalized linear mixed-effects models (GLMMs) used to describe the response of the dolphin groups (avoidance) to the presence of vessels in the observation area (50 to 300 m around the group). The tilde (~) is read as “in function of”.

Model Rank	Model Description (Avoidance Response~)	k	AIC_c_	∆AIC_c_	*W_i_*
1	Vessels + compliance + size + cohesion	7	143.58	0.00	0.76
2	Vessels + compliance + size	5	147.33	3.75	0.12
3	Behaviour + vessels + compliance + cohesion + size	11	148.57	4.99	0.06
4	Behaviour + vessels + compliance + cohesion	10	149.91	6.33	0.03
5	Vessels + compliance	4	151.43	7.85	0.02
6	Behaviour + vessels + compliance + size	9	152.92	9.34	0.01
7	Compliance + cohesion	5	155.03	11.45	0.00
8	Compliance + cohesion + size	6	155.86	12.28	0.00
9	Behaviour + vessels + compliance	8	156.15	12.57	0.00
10	Behaviour + compliance + cohesion	9	158.87	15.28	0.00
11	Compliance + size	4	159.52	15.94	0.00
12	Compliance	3	160.17	16.59	0.00
13	Behaviour + compliance + size + cohesion	10	160.26	16.68	0.00
14	Behaviour + compliance	7	164.72	21.13	0.00
15	Behaviour + compliance + size	9	164.89	21.31	0.00

Notes: The full model was avoidance ~ behaviour + vessels + non-compliance/compliance + cohesion + size + (1|focal group). All models included a random factor: + (1|focal group). Model descriptions are shown with the number of parameters (k), AICc, difference in AICc from the best-supported model (∆AICc) and AICc weights (*Wi*). Only the first 15 best models are included in this table.

## Data Availability

The data presented in this study are available on request from the corresponding author.

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
