# Peer review of "Dolphin Watching and Compliance to Guidelines Affect Spinner Dolphins’ (*Stenella longirostris*) Behaviour in Reunion Island"

_animals, 2021, doi:10.3390/ani11092674_

Round 1
Reviewer 1 Report
This paper aims to assess the effect of dolphin-watching on spinner dolphins through focal follows. This article is interesting but the study could be substantially improved.
My main concern in this study is the potential influence of the uneven sampling effort (larger dataset in the impact condition than in the control condition). After seeing the Table 2, I would suggest to randomly select the same number of samples both in the impact condition and in the control condition.
My second concern is regarding the use of a vessel to study the potential influence of dolphin-watching boats on dolphin behaviour. All the data in the control condition were collected from a vessel, which could also potentially affect the behaviour of the dolphins. Can the authors justify the use of a research vessel (rather than a land-based platform for example)? Can the authors explain why the dolphin-watching boats would be the only ones affecting dolphins’ behaviour, and not the research vessel?
Such pitfalls make the discussion is speculative.
The introduction is rather short and a more detailed literature review would benefit the manuscript.
On a smaller note, authors should be consistent with the use of the term “group size” and not switching randomly with the word “pod”. In cetaceans, a pod is considered a group of animals that do not change in composition over time. I would suggest to keep “group size” throughout the full manuscript.
Reviewer 2 Report
Dolphin-watching and compliance to guidelines affect spinner dolphins’ behaviour (Stenella longirostris) in Reunion Island
This is an interesting article on how non-compliance to dolphin-watching guidelines can effect the behaviour of spinner dolphins in the Reunion Islands. I think the writing in the manuscript should be tightened before it can be considered for publication. I also had some concerns regarding the behaviour definitions of the spinner dolphins and why behaviours were based on other dolphin species and not spinner dolphins? I am also concerned on how the distances of the vessels to the focal dolphin groups were measured. There is more detail below.
Introduction
Line 85 – 86 should add ‘at night’ after foraging.
Lines 94 it might be beneficial to define what a ‘Prefectural Order’ is
Methods
2.1 Study Area
Figure 1 caption should be directly below the figure of the map
2.2 Data collection
Why did the authors choose a 300-metre radius?
How did the authors determine that the research vessel was 300 m or more from the focal group of spinner dolphins? Was the measurement taken from the middle or edge of the dolphin group? Why did was the vessel positioned parallel to the focal group and not behind? Do the authors think that being behind the group would have had less disturbance on the focal group rather than being parallel? Did the authors consider using a land-based theodolite to collect the spinner dolphin behavioural state data, and the vessel data?
2.2.1 Group composition and behavioural state
Why did the authors take the behavioural state definitions from non-spinner dolphin behaviours, rather than spinner dolphin behavioural states such as those defined by Norris et al 1994 in the Hawaiian spinner dolphin and Norris and Dohl 1980 Behaviour of the Hawaiian Spinner Dolphin? One of the main characteristics of the spinner dolphin resting behaviour is that they spend more time submerged (1.5 – 3mins) which isn’t in the authors definition, and also reduce their aerial behaviours and acoustic activity. Do the authors think that the difference in the behavioural state definitions will have an effect on their analysis of the spinner dolphins behavioural states during both the control and impact conditions? Did the authors consider all behaviour of all age groups during the analysis or just adult animal behaviours? How did the authors handle fission and fusion situations, so what was the protocol if the focal group split or if the focal group was joined by another group during the focal follow?
2.2.2 Dolphin behavioural response to vessels
How did the authors measure the distance of vessels to the focal dolphin group? Did the authors measure the response of all animal age classes in the focal group to vessels, or was it just adults?
2.2.3 Vessels compliance with regulations
How did the authors measure the distance from the vessels to the focal dolphin group?
Results
I think it would be good to have a summary table that shows the number of focal follows, mean group size, mean focal follow duration etc. This will help put the results into context.
Is there any information on the abundance and survival rates of this spinner dolphin population?
Reviewer 3 Report
This is in my view a very nicely written manuscript about a fine research study on an important and timely topic. The ms is specific and detailed, although it also discusses important general concerns about cetacean-watching. The cited references are thorough and up to date, the methods of data collection and analysis are reasonable, and the figures and tables are useful.
One of the questions I had was whether the observations (from February 2018 to June 2020, 7 am to 2 pm) coincided with any major events, such as seasonal presence of prey, that might have affected dolphin behavior. Yes, I know the authors wrote that they observed no feeding behavior, and it is good that they included such notes, but I want to be able to rule out any other phenomena that might potentially have affected dolphin behavior, such as the presence of prey, predators, or competitors, or seasonal currents, and so on.
It is good that all of the categorized behaviors are mutually exclusive. Is it known from other studies (either of spinner dolphins or other cetaceans) if one type of behavior normally follows other specific behaviors—for example, that travelling is typically followed by resting or milling, or if socializing generally precedes diving (again, not necessarily for the animals in this study, but in general for say sperm whales or other odontocetes).
The information on vessel compliance and on dolphin recovery times are interesting.
Maybe I missed it, but did the authors state how many dolphin-watching vessels usually are found on the west coast of Reunion? (Are we talking two boats or many more? Just wondering…) Do they go out year-round? How large are these vessels? How many passengers?
A few very minor suggestions:
Table 1: Hard to read. Put lines between categories, as in Table 2
In Table 3, there is enough room to state the type of formation (2=Dispersed) within the table itself, not just in the caption.
Line 13: over (not at) short and long terms?
Line 124: allowed researchers to…? (missing a word here)
Round 2
Reviewer 1 Report
The authors choose not to do follow the reviewer suggestion regarding the random selection of both control and impact samples. The results of this study might therefore be seriously biased. The methodology is not adequate to be published in a peer-reviewed journal. I therefore recommend rejection.
Reviewer 3 Report
I thank the authors for carefully considering my comments and suggestions. Their revisions have improved this manuscript, which I think will be a nice contribution to the published literature.
